# Flurona: The First Autopsied Case

**DOI:** 10.3390/medicina59091616

**Published:** 2023-09-07

**Authors:** Ionuț Isaia Jeican, Dan Gheban, Alexandra Mariș, Silviu Albu, Maria Aluaș, Costel Vasile Siserman, Bogdan Alexandru Gheban

**Affiliations:** 1Department of Anatomy and Embryology, Iuliu Hatieganu University of Medicine and Pharmacy, 400006 Cluj-Napoca, Romania; 2Department of Pathology, Iuliu Hatieganu University of Medicine and Pharmacy, 400006 Cluj-Napoca, Romania; 3Department of Pathology, Emergency Clinical Hospital for Children, 400370 Cluj-Napoca, Romania; 4Intensive Care Unit, Emergency Clinical Hospital for Children, 400370 Cluj-Napoca, Romania; alexandramaris85@gmail.com; 5Department of Head and Neck Surgery and Otorhinolaryngology, University Clinical Hospital of Railway Company, Iuliu Hatieganu University of Medicine and Pharmacy, 400015 Cluj-Napoca, Romania; 6Department of Oral Health, Iuliu Hatieganu University of Medicine and Pharmacy, 400012 Cluj-Napoca, Romania; 7Institute of Legal Medicine, 400006 Cluj-Napoca, Romania; 8Department of Legal Medicine, Iuliu Hatieganu University of Medicine and Pharmacy, 400006 Cluj-Napoca, Romania; 9Department of Histology, Iuliu Hatieganu University of Medicine and Pharmacy, 400349 Cluj-Napoca, Romania; 10Department of Pathology, Emergency Clinical County Hospital, 400347 Cluj-Napoca, Romania

**Keywords:** flurona, COVID-19, influenza, autopsy

## Abstract

COVID-19-associated coinfections increase the patient’s risk of developing a severe form of the disease and, consequently, the risk of death. The term “flurona” was proposed to describe the coinfection of the influenza virus and SARS-CoV-2. This report is about a case of a 7-month-old female infant who died due to flurona coinfection. A histopathological exam showed activation of microglia (becoming CD45 positive), bronchial inflammation, diffuse alveolar damage in proliferative phase with vasculitis, a peribronchial infiltrate that was predominantly CD20-positive, and a vascular wall infiltrate that was predominantly CD3-positive. The aggressiveness of the two respiratory viruses added up and they caused extensive lung inflammation, which led to respiratory failure, multiple organ failure, and death. Tissues injuries caused by both the influenza virus and SARS-CoV-2 could be observed, without the ability to certify the dominance of the aggression of one of the two viruses.

## 1. Introduction

COronaVIrus Disease 19 (COVID-19)-associated coinfections represent both a diagnostic and a therapeutic challenge. Bacterial and fungal coinfections have been reported by numerous studies, both in association with COVID-19 [1,2,3,4,5] and with the flu [6]. Coinfections increase the patient’s risk of developing a severe form of the disease and, consequently, the risk of death. As epidemiological restrictions are lifted, respiratory coinfections are expected in the coming winters, a hypothesis that emphasizes the importance of seasonal vaccination [7].

Severe acute respiratory syndrome coronavirus 2 (SARS-CoV-2) and influenza virus coinfection could have a significant impact on morbidity, mortality, and health service demand. Until now, there are very limited data regarding this coinfection. It seems that although this coinfection affects approximately 0.8% of patients with COVID-19, the risk of death is nearly six times greater [8]. A study performed in Great Britain on 6965 COVID-19 patients showed that viral coinfection was detected in 583 patients, out of which 227 patients had influenza virus coinfection (3.25%) [7]. The proportion of SARS-CoV-2 and influenza coinfection can vary from country to country because of the seasonal pattern of influenza [9].

The proportion of coinfection with influenza virus and SARS-CoV-2 among children seems to be higher than that in adult patients, suggesting that children are more susceptible to the coinfection. The proportion of coinfection with influenza viruses among critically ill COVID-19 patients seems to also be higher than that in overall patients, suggesting that coinfection with influenza viruses may aggravate the severity of COVID-19 [9]. The reported figures may increase in the future, given the higher infectivity of new SARS-CoV-2 strains [10,11] and the continued co-circulation of SARS-CoV-2 and influenza viruses [12].

In January 2022, the term “flurona” was proposed to describe the coinfection of the influenza virus and SARS-CoV-2 [13], which was preferred over the other proposed term “flucovid” [14]. 

In this study, we followed a study of histopathological aspects and their correlation with the death of a patient with SARS-CoV-2 and influenza A virus coinfection. To the best of our knowledge, this is the first paper reporting an autopsy of a case where flurona resulted in death.

## 2. Materials and Methods

**Ethical and legal considerations**. Our study was conducted by considering the provisions of the Romanian legal frameworks (Law 104/2003 on the handling of bodies and the removal of organs and tissues with a view to transplantation, as well as Government Decision No. 451/2004 on methodological norms for the application of Law 104/2003) and specific international and national recommendations for COVID-19 [15,16,17,18]. The harvesting protocol for this study was approved by the Ethics Committee of Iuliu Hatieganu University of Medicine and Pharmacy, Cluj-Napoca, Romania, No. 63/01.03.2022.

**Preparation of samples for the histopathological examination**. The tissue samples collected during the autopsy were fixed in 7% formaldehyde for 5 days, after which the samples were oriented and placed in cassettes. Tissue processing was performed using a vacuum infiltration processor, namely a Tissue-Tek VIP 5 Jr (Sakura, Alphen aan den Rijn, Netherlands). Paraffin embedding and sectioning were performed using the Tissue-Tek TEC 6 system (Sakura, Alphen aan den Rijn, Netherlands) and Accu-Cut SRM 200 Rotary Microtome (Sakura, Alphen aan den Rijn, Netherlands). Slide staining was performed using the automated slide stainer Tissue-Tek Prisma Plus (Sakura, Alphen aan den Rijn, Netherlands) according to the internal staining protocol using Mayer Modified Hematoxylin (Titolchimica, Rovigo, Italy) and Eosine solution (10 g Eosine B in 1000 mL distilled water). For Gram staining, the Gram Stain Kit (Gram Fuchsin Counterstain) (Atom Scientific, Manchester, UK) was used.

**Preparation of samples for immunohistochemistry**. Immunohistochemistry was performed automatically on 3 μm thick sections of formalin-fixed and paraffin-embedded tissues with MD Stainer (Vitro Master Diagnostica^®^, Granada, Spain). For the immunohistochemical assessment, we used Agilent Dako antibodies (Santa Clara, CA, USA) LCA (clone 2B11 + PD7/26) at a 2:100 dilution, CD3 (clone F7.2.38) at a 2:100 dilution, and CD20 (clone L26) at a 0.25:100 dilution.

**Microscopic examination**. Microscopic examination was performed by an experienced pathologist (D.G.) using a Leica DM1000 clinical microscope (Leica, Wetzlar, Germany) with a dedicated image acquisition camera and software. All sections were examined by the same experienced investigator (D.G.).

## 3. Case Description

**Clinical presentation**. We present the case of a 7-month-old female infant who was admitted, on 13 January 2023, to the hospital for an unresponsive recurrent fever (up to 39.2 °C) after the administration of paracetamol. At the physical examination she weighed 7650 g and was 70 cm tall, with altered general condition, fever, dysphonia, bilaterally intensified vesicular murmur, and bilateral bronchial rales observed. Oxygen saturation (SaO_2_) ranged from 90 to 92% in ambient air. Laboratory investigations revealed a marked inflammatory syndrome, and rapid tests for COVID-19 and influenza type A and B yielded positive results for COVID-19. A chest X-ray revealed a pattern consistent with right-sided pneumonia. Injectable antibiotic treatment with Ceftriaxone, Gentamicin, Hydrocortisone Hemisuccinate, Miofilin, probiotics, and oxygen therapy through a facial mask was initiated. 

After starting treatment administration, the patient’s condition showed a slightly favorable evolution. However, after 3 days of hospitalization, the patient displayed marked respiratory distress, with desaturation down to 80–82% under oxygen therapy using a mask at a flow rate of 12 L/min, leading to the initiation of Bubble CPAP non-invasive ventilation. The pulmonary X-ray revealed a worsened appearance of pneumonia (diffuse congestive focus extended bilaterally in the perihilar region and the right hiliobasal region). The Multiplex PCR test performed on nasopharyngeal secretion yielded positive results for both SARS-CoV-2 and influenza virus type A. On the same day, the patient was transferred to the ICU, where she presented with a body temperature of 37.2 °C, pale, warm, dry skin and mucous membranes, reduced skin turgor, a heart rate of 121 beats per minute, BP = 118/67 mmHg, regular heart sounds without murmurs, ample peripheral and central pulses, marked respiratory effort, intensified vesicular murmur with scattered bronchial rales bilaterally, a distended abdomen without signs of guarding, hepatomegaly 2 cm below the costal margin, and pronounced psychomotor agitation. In terms of laboratory results, the patient exhibited leukocytosis and neutrophilia (leukocytes = 38,170/mm^3^, neutrophils = 22,000/mm^3^ (57.8%)) despite the antibiotic treatment, a bacterial infection that was probably superimposed. A slight lymphopenia (3500/mm^3^ (34.3%)) was likely due to viral infections. The laboratory results also indicated a moderate normochromic normocytic anemia, no coagulation disorders, mild cytolysis syndrome (ASAT = 200 U/L, ALAT = 77 U/L) that was probably damage to the liver resulting from drugs and/or direct viral aggression (see discussions), and minimal inflammatory syndrome (CRP = 1.54 mg/dL, PCT = 1.74 ng/ mL).

The following treatment plan was initiated: high-flow nasal oxygen therapy, hydro-electrolytic rehydration, broad-spectrum antibiotic therapy (Cefotaxime and Teicoplanin), intravenous Remdesivir and oral Tamiflu administration, anticoagulant treatment with low-molecular-weight heparin, vitamin therapy (C, D, B), respiratory physiotherapy and kinetotherapy, and aerosol therapy.

Approximately 5 h later, the patient presented a picture of aggravated acute respiratory failure involving increased psychomotor agitation, and she experienced a sudden desaturation episode despite increasing the high-flow parameters to maximum values based on weight and age and exhibited marked respiratory effort with intercostal retractions and Astrup parameters with severe decompensated mixed acidosis (hypercapnic, metabolic, lactic) (ph = 7.03, pCO_2_ of 99 mmHg, lac = 12 mmol/L, HCO_3_ = 18 mEq/L). A decision was made for rapid sequence oro-tracheal intubation, but balloon ventilation failed and the patient presented spasticity and thoracic stiffness, along with cervical and upper thoracic subcutaneous emphysema. An emergency chest X-ray revealed bilateral tension pneumothorax, explainable in the context of severe bilateral lung damage. Two chest tubes were inserted, leading to partial re-expansion of the right lung followed by the resorption of subcutaneous emphysema.

During the patient’s progression, her hemodynamic stability deteriorated. A decision was made to insert a central venous catheter into the right femoral vein, and continuous therapy with norepinephrine was initiated. After 5 days of hospitalization, the patient’s condition worsened with aggravated Astrup parameters (persistent metabolic and lactic acidosis with increasing serum lactate levels, despite an improvement in CO_2_ values). Marked hemodynamic instability necessitated escalation of norepinephrine administration to maximum levels and introduction of the use of adrenaline. The patient went into cardiac arrest, and pharmacological and mechanical resuscitation maneuvers were performed. After 30 min, a sinus rhythm was restored. An hour later, the patient experienced another cardiac arrest, and resuscitation was successful within 9 min, resulting in a return to a sinus rhythm. After an additional hour, given the severely compromised state, ongoing hemodynamic instability, and persistent metabolic acidosis, the patient entered irrecoverably into cardiac arrest.

**Macroscopic autopsy results**. After 12 h from death, a pathological autopsy was performed. Macroscopically, the following observations were made: leptomeningeal congestion (Figure 1A), cerebral edema with tonsillar herniation (Figure 1B, arrows), pneumomediastinum, bilateral pneumothorax (Figure 1C), bullous emphysema, bilateral lung consolidation (Figure 1D), mild left ventricular hypertrophy, mild aortic coarctation (Figure 1E), hepatomegaly (Figure 1F), shock kidneys (Figure 1G), and normal appearance of the thymus and lymph nodes (Figure 1H).

**Histopathological and immunohistochemical results**. Histopathologically, in the brain, cerebral congestion was observed (Figure 2A), along with extensive cerebral edema (Figure 2B). Hypoxic encephalopathy was revealed by multifocal neuronal necrosis, Purkinje cell necrosis (Figure 2C), and cerebritis with micronecrosis in the olfactory bulb (Figure 2D,E). At the pulmonary level, alveolar collapse and subpleural interstitial emphysema were noted (Figure 3A), along with congestion and mucus plugging in bronchioles (Figure 3B) that extended into alveoli (Figure 3C). There was marked bronchial inflammation (Figure 3D), diffuse alveolar damage (DAD) with hyaline membranes populated by fibroblasts (proliferative phase) (Figure 3E), alveolar congestion and intense interstitial inflammation (Figure 3F), and vasculitis (Figure 3G,H). The liver presented acute lobular hepatitis with moderate hydropic hepatocellular degeneration and Councilman bodies in sinusoids (Figure 4A), while the kidney showed proximal tubular necrosis and distal tubular necrobiosis (Figure 4B).

The immunohistochemical examination revealed, in the olfactory bulb, positive staining for microglial CD45 (LCA) cells (Figure 2F), while CD3 and CD20 staining was negative. At the pulmonary level, the peribronchial infiltrate and the one in the vascular wall showed CD45 (LCA) positivity (Figure 3J). The peribronchial infiltrate was predominantly CD20-positive (Figure 3K), whereas the infiltrate in the vascular wall was predominantly CD3-positive (Figure 3L).

## 4. Discussion

It is known that pediatric patients experience significant morbidity due to influenza and are considered key factors in virus transmission, while young children infected with SARS-CoV-2 appear to be generally asymptomatic or only exhibit mild symptoms [12].

In the case presented by us, the initial lung damage (right pneumonia) spread bilaterally quickly (3 days) and led to a respiratory failure that got progressively worse, despite intensive treatment. Thus, we find that the treatment was ineffective in the face of double viral aggression, which produced persistent lung inflammation. Histopathologically, the interstitial pulmonary inflammation (Figure 3F) was much more intense compared to the autopsied and analyzed COVID-19 cases by our team during the pandemic period (over 200 cases, data still unpublished). Animal studies have shown that simultaneous coinfection by influenza A virus and SARS-CoV-2 resulted in more severe lung inflammatory damage due to increased tissue cytokine/chemokine expression than infection by either influenza A virus or SARS-CoV-2 alone [19].

Also, at the pulmonary level, we noted very intense vasculitis (Figure 3G,I), with a rich infiltrate of T lymphocytes (Figure 3L) without associated thrombosis. Thus, we can emphasize double lung damage, both perfusion and diffusion. Vascular involvement in influenza A H1N1 and COVID-19 was compared in one previous autopsy study. The results showed that for both patient categories, the histologic pattern in the peripheral lung was diffuse alveolar damage with perivascular T-cell infiltration. Also, the lungs of COVID-19 patients showed severe endothelial injury, and alveolar–capillary micro thrombi were nine times as prevalent in patients with COVID-19 as in patients with influenza [20]. While infiltration of over-activated cytotoxic T cells can be harmful in the infected tissue, fast-responding T cells are important in the protection against severe COVID-19 [21]. T-cell responses develop early and correlate with protection but are relatively impaired in severe forms of disease and are associated with intense activation and lymphopenia [22].

Being rich in B lymphocytes (Figure 3K), we consider that bronchitis/bronchiolitis is likely due to the influenza virus. Additionally, in our previous studies, we observed that COVID-19 infection generally does not produce intense histological inflammation in the bronchial structures [23,24]. The involvement of bronchial structures has been reported in infections with influenza, adenovirus, herpes, varicella-zoster, and respiratory syncytial virus [25]. The characteristic plugs of viscous mucus in bronchiolitis (a specific reaction pattern in infants) significantly worsened the progression, likely contributing to the development of spontaneous bilateral tension pneumothorax. Thus, the aggression of both viruses on the lung can be argued.

The macroscopic and microscopic changes which we found at the cerebral level are due to hypoxic encephalopathy secondary to persistent respiratory failure. The studies show that the encephalopathic manifestations of COVID-19 are mainly due to clinically significant hypoxia [26]. We have not encountered micronecrosis in the olfactory bulbs of COVID-19 patients in previous studies. The LCA staining (Figure 2F) revealed that what appeared as lymphocytes in the hematoxylin–eosin staining (Figure 2D), which were interpreted as cerebritis, was actually microglial cells that became positive for LCA when activated [27,28,29], with CD3 and CD20 staining being negative.

Direct viral aggression on the liver can be argued by the presence of Councilman bodies (Figure 4A), their presence being characteristic in different viral syndromes [30,31]. Renal necrosis (Figure 4B) additionally argues for multiple organ failure.

## 5. Conclusions

The aggressiveness of the two respiratory viruses added up and caused extensive lung inflammation. The pathogenetic chain that led to the death of the patient was initiated by persistent pulmonary inflammation, unresponsive to treatment, which affected pulmonary diffusion (proliferative DAD, bronchiolitis) and perfusion (pulmonary vasculitis). This caused respiratory failure, followed by multiple organ failure and death. Histopathologically, tissue injuries caused by both the influenza virus and SARS-CoV-2 could be observed, without the ability to certify the dominance of the aggression of one of the two viruses.

The fact that the pulmonary inflammation was so persistent and progressed, despite treatment, to the proliferative phase indicates the need for further research to figure out the optimal treatment of this coinfection.

## Figures and Tables

**Figure 1 medicina-59-01616-f001:**
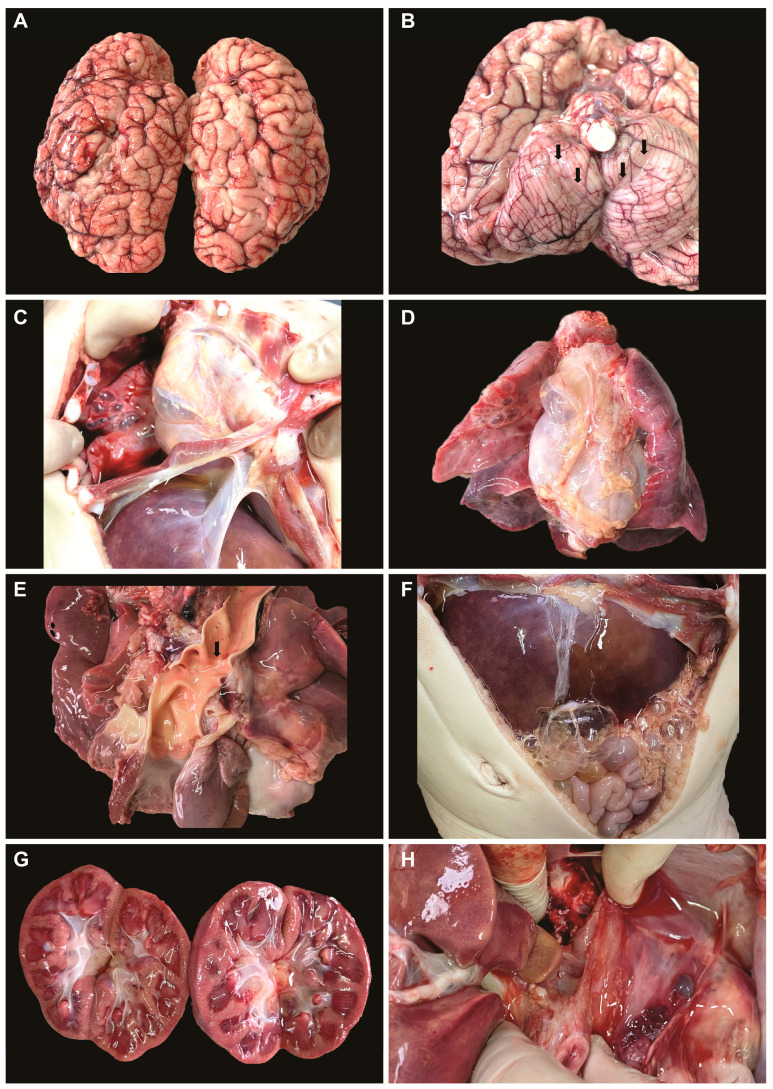
Macroscopic observations during autopsy: (**A**) leptomeningeal congestion; (**B**) cerebral edema with tonsillar herniation; (**C**) pneumothorax; (**D**) bullous emphysema, lung consolidation; (**E**) left ventricular hypertrophy, mild aortic coarctation; (**F**) hepatomegaly; (**G**) shock kidneys; (**H**) normal appearance of the thymus.

**Figure 2 medicina-59-01616-f002:**
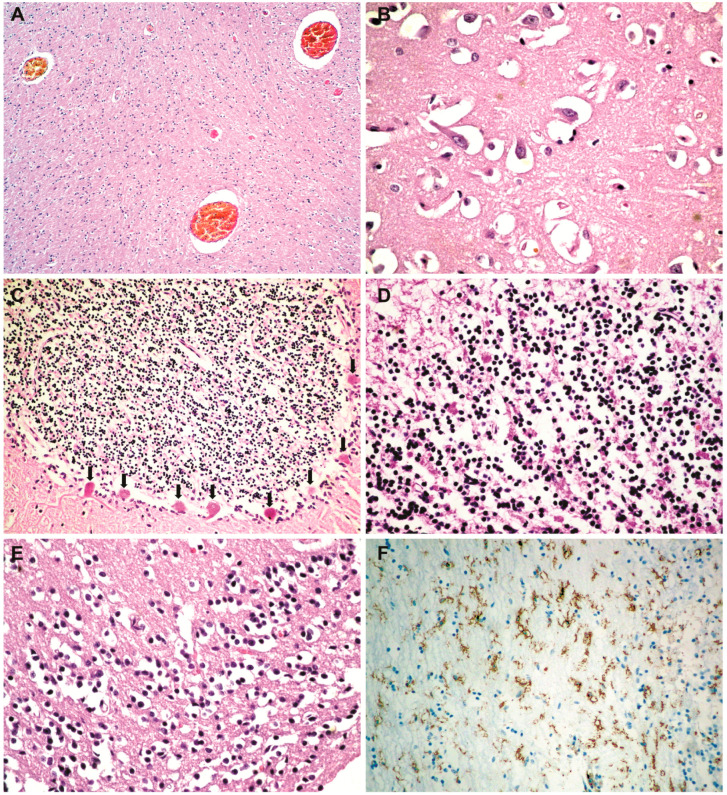
Cerebral histopathology and immunohistochemistry: (**A**) cerebral congestion (HE × 40); (**B**) extensive cerebral edema (HE × 400); (**C**) Purkinje cell necrosis (hypoxic encephalopathy) (HE × 100); (**D**) cerebritis with micronecrosis in the olfactory bulb (HE × 200); (**E**) micronecrosis in the olfactory bulb (HE × 200); (**F**) microglial CD45-positive cells in the olfactory bulb (LCA × 100).

**Figure 3 medicina-59-01616-f003:**
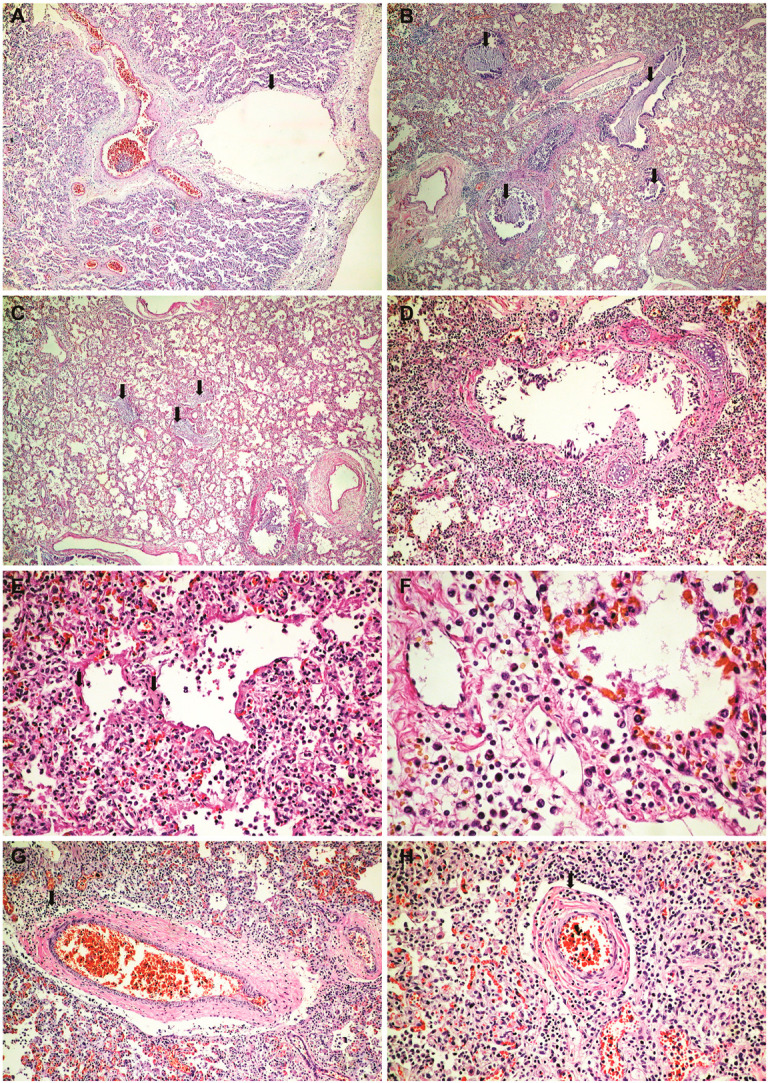
Lung histopathology and immunohistochemistry: (**A**) alveolar collapse and subpleural interstitial emphysema bubble (arrow) (HE × 40); (**B**) congestion and mucus plugging in bronchioles (HE × 40); (**C**) extension of mucus plugging into alveoli (HE × 40); (**D**) intense bronchial inflammation (HE × 100); (**E**) diffuse alveolar damage (DAD) with hyaline membranes populated by fibroblasts (proliferative phase) (HE × 200); (**F**) alveolar congestion and intense interstitial inflammation (HE × 400); (**G**–**I**) vasculitis (HE × 100); (**J**) a peribronchial infiltrate and vascular wall infiltrate showing CD45 positivity (LCA × 100); (**K**) intense positive CD20 staining in the peribronchial infiltrate (CD20 × 40); (**L**) intense positive CD3 staining in the vascular wall infiltrates (CD3 × 100).

**Figure 4 medicina-59-01616-f004:**
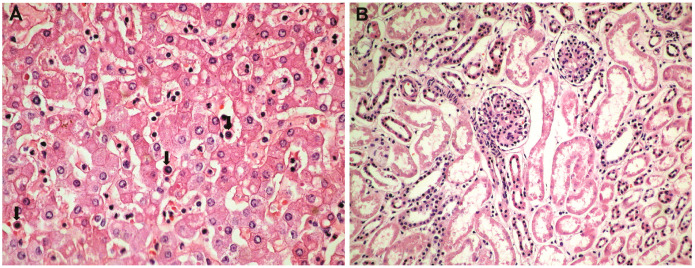
Hepatic and renal histopathology: (**A**) acute lobular hepatitis with moderate hydropic hepatocellular degeneration and Councilman bodies (arrows) in sinusoids (HE × 100); (**B**) proximal tubular necrosis and distal tubular necrobiosis (HE × 100).

## Data Availability

The autopsy results are available at the Department of Pathology, Emergency Clinical Hospital for Children, 400370 Cluj-Napoca, Romania; dan.gheban@umfcluj.ro.

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
