# Peer review of "Flurona: The First Autopsied Case"

_medicina, 2023, doi:10.3390/medicina59091616_

Round 1
Reviewer 1 Report
an autopsy report must report the weight of different organs in a table associated to the lenght and to estimated weight of the child
herein these data are missing, therefore I advise the authors to include them i the paper
quality of English is acceptable
Author Response
Dear Reviewer, thank you very much for the comment and suggestions! Data on weight of organs are not available because the local autopsy protocol does not provide for organ weighing in patients under 3 years of age. However, at the moment of the admission to the hospital, the patient weighed 7650 g and was 70 cm tall, information added to the manuscript (see line 102).
Reviewer 2 Report
While potentially outside of the scope of research, it would be very interesting to stain with IHC for SARS-CoV2 & IAV proteins to see where the viruses are present and if they are coinfecting the same cells, or just some tissues.
Author Response
Dear Reviewer, Thank you very much for your observation and suggestions! Your research idea is very interesting. In the time available for the review changes, we did not find IAV antibodies for IHC in Romania. We will consider, for sure, your proposal in further research on co-infections.Reviewer 3 Report
Abstract: There are some issues that need attention. Firstly, the term "flurona" is introduced without sufficient context or definition in the abstract. While it's mentioned that the patient died due to flurona coinfection, the abstract should provide more information about what "flurona" means and why it is significant. Furthermore, the list of histopathological findings in the autopsy is quite extensive and might be overwhelming for readers in the abstract. It would be more effective to focus on the most critical findings and their implications for the patient's death.
Introduction: The introduction is not well-structured, and its flow is disrupted by the abrupt introduction of the term "flurona." The explanation and significance of "flurona" should be expanded here rather than leaving readers to infer its meaning from the rest of the text. Additionally, while the study's focus on a specific case is appropriate, the introduction could benefit from a clearer statement of the study's objective or research question. The references cited could be elaborated upon to provide a more comprehensive understanding of the existing literature on coinfections.
Materials and Methods: The materials and methods lacks clarity in terms of its organization and could be divided into subsections for better readability. The ethical considerations and approval process should be presented before the technical details of the tissue processing and staining procedures. Additionally, while the specific methods for tissue processing and staining are provided, there is no explanation of why these methods were chosen or how they contribute to the study's goals. Adding a brief rationale for these methods would improve the section.
Case Description: This section is well-detailed and provides a comprehensive overview of the patient's clinical course. However, the narrative could be improved by organizing the information more systematically, possibly in a chronological order of events. This would make it easier for readers to follow the progression of the case. Additionally, the section could benefit from more critical analysis and discussion of the clinical findings and their implications. For instance, rather than just presenting the laboratory results, discussing their significance in the context of the patient's deteriorating condition would enhance the understanding of the case.
Discussions: The discussions section lacks proper structure and coherence. The authors jump between different topics and studies without clear transitions. The discussion should be organized according to the major findings of the case and their implications. Each finding should be discussed in detail, referring to relevant literature for comparison and interpretation. The section also lacks a clear conclusion or summary of the discussions.
Conclusions: The conclusions section lacks depth. Instead of merely restating that the patient's death was attributed to coinfection, the authors should provide a more thorough synthesis of the findings and their implications. The conclusion should address the research question or objectives stated in the introduction and offer insights into the broader implications of the case for understanding the interaction between COVID-19 and influenza coinfections.
Overall, the manuscript needs improvement in terms of organization, clarity, and critical analysis. By addressing the shortcomings outlined above, the manuscript can be more effectively presented and its findings more comprehensively discussed.
Here are some of the grammatical and English mistakes found in the manuscript:
Abstract:
- "This report is about a case of a 7-month-old female infant who was deceased due to flurona coinfection." - Instead of "was deceased," use "died" or "passed away."
- "Autopsy performed showed multifocal neuronal necrosis, activation of microglia (become CD45 positive), bronchial inflammation..." - Change "become" to "becoming."
Introduction: 3. "COVID-19-associated coinfections represent both a diagnostic and a therapeutic 35 challenge." - There seems to be a number "35" that's not integrated correctly into the sentence.
- "As epidemiological restrictions are lifted, respiratory coinfections are expected in 39 the coming winters..." - Consider rephrasing as "As epidemiological restrictions are lifted, respiratory coinfections are expected in the coming winters..."
- "In January 2022, the term 'flurona' was proposed to describe the coinfection of the 42 influenza virus and SARS‐CoV‐2 [5]." - The numbering seems to be mixed with the text, as "42" appears between "the" and "influenza."
- "The continued co‐circulation of SARS‐CoV‐2 and influenza viruses is ex‐ 49 pected to present challenges [7]." - The numbering appears between "ex" and "pected."
- "To the best of our knowledge, this is the first paper re- 50 porting an autopsy of a case deceased by flurona." - "re- 50 porting" should be "reporting."
Materials and Methods: 8. "Our study was conducted by considering the provisions of the Romanian legal 53 frameworks..." - "legal 53 frameworks" should be "legal frameworks" without the number.
- "...and Pharmacy, Cluj-Napoca, Romania, No. 63/1.03.2022." - "No. 63/1.03.2022" should be "No. 63/01.03.2022" for consistency.
- "The tissue samples collected during the autopsy were fixed in formaldehyde 7% for 60 days..." - "formaldehyde 7%" should be "7% formaldehyde."
- "Paraffin embedding and sectioning were performed 63 using the Tissue-Tek TEC 6 system..." - Change "63 using" to "using" for correct syntax.
Case Description: 12. "...fever, dysphonia, bilaterally intensified vesicular murmur, and bilateral bronchial 85 rales are observed." - It seems the numbering "85" is embedded in the sentence.
- "Approximately 5 hours later, the patient displays increased psychomotor agitation, 116 she experiences a sudden desaturation episode..." - The numbering "116" is within the sentence.
- "...intercostal retractions, Astrup parameters with severe decompensated mixed aci- 119 dosis (hypercapnic, metabolic, lactic) (ph=7.03, pCO2 of 99 mmHg, lac=12mmol/L, 120 HCO3=18 mEq/L)." - The numbering "119 dosis" is not properly separated.
- "The characteristic plugs of viscous mucus in bronchiolitis (a specific re-action pattern in infants) significantly worsened the progression, likely contributing to 225 the development of spontaneous bilateral tension pneumothorax." - The numbering "225" is misplaced.
Discussions: 16. "Being rich in B lymphocytes (fig. 3K), we consider that bronchitis/bronchiolitis is 219 likely due to the influenza virus." - Change "219 likely" to "likely 219."
Conclusions: 17. "Histopathological, 236 tissues injuries caused by both the influenza virus and SARS-CoV-2 can be observed." - "tissues injuries" should be "tissue injuries."
These are some of the grammatical and English mistakes found throughout the manuscript. Please note that there may be additional errors that were not highlighted in this review.
Author Response
Dear Reviewer, many thanks for your comments, opinions and suggestion on this manuscript. Here are our responses to your observations.
Q1: Abstract: There are some issues that need attention. Firstly, the term "flurona" is introduced without sufficient context or definition in the abstract. While it's mentioned that the patient died due to flurona coinfection, the abstract should provide more information about what "flurona" means and why it is significant. Furthermore, the list of histopathological findings in the autopsy is quite extensive and might be overwhelming for readers in the abstract. It would be more effective to focus on the most critical findings and their implications for the patient's death.
Response: Thank you for suggestions! The abstract was completed taking into consideration your suggestions.
Q2: Introduction: The introduction is not well-structured, and its flow is disrupted by the abrupt introduction of the term "flurona." The explanation and significance of "flurona" should be expanded here rather than leaving readers to infer its meaning from the rest of the text. Additionally, while the study's focus on a specific case is appropriate, the introduction could benefit from a clearer statement of the study's objective or research question. The references cited could be elaborated upon to provide a more comprehensive understanding of the existing literature on coinfections.
Response: The introduction was completed according to your suggestions. See lines 44-65.
Q3: Materials and Methods: The materials and methods lacks clarity in terms of its organization and could be divided into subsections for better readability. The ethical considerations and approval process should be presented before the technical details of the tissue processing and staining procedures. Additionally, while the specific methods for tissue processing and staining are provided, there is no explanation of why these methods were chosen or how they contribute to the study's goals. Adding a brief rationale for these methods would improve the section.
Response: The section was completed as you suggested. We cannot explain more about these methods, they are standard methods used in pathology laboratories.
Q4: Case Description: This section is well-detailed and provides a comprehensive overview of the patient's clinical course. However, the narrative could be improved by organizing the information more systematically, possibly in a chronological order of events. This would make it easier for readers to follow the progression of the case. Additionally, the section could benefit from more critical analysis and discussion of the clinical findings and their implications. For instance, rather than just presenting the laboratory results, discussing their significance in the context of the patient's deteriorating condition would enhance the understanding of the case.
Response: The section was completed as you suggested.
Q5: Discussions: The discussions section lacks proper structure and coherence. The authors jump between different topics and studies without clear transitions. The discussion should be organized according to the major findings of the case and their implications. Each finding should be discussed in detail, referring to relevant literature for comparison and interpretation. The section also lacks a clear conclusion or summary of the discussions.
Response: The Discussion part was completed according to your suggestions. See lines 212-257.
Q6: Conclusions: The conclusions section lacks depth. Instead of merely restating that the patient's death was attributed to coinfection, the authors should provide a more thorough synthesis of the findings and their implications. The conclusion should address the research question or objectives stated in the introduction and offer insights into the broader implications of the case for understanding the interaction between COVID-19 and influenza coinfections.
Response: The Conclusion section was completed as you requested.
Q7: Overall, the manuscript needs improvement in terms of organization, clarity, and critical analysis. By addressing the shortcomings outlined above, the manuscript can be more effectively presented and its findings more comprehensively discussed.
Response: Thank you!
Comments on the Quality of English Language
Here are some of the grammatical and English mistakes found in the manuscript:
Response: Thank you for your helpful observations! We have done all English changes you suggested, to improve the manuscript. All changes we have done are highlighted on yellow in the manuscript.
Abstract:
- "This report is about a case of a 7-month-old female infant who was deceased due to flurona coinfection." - Instead of "was deceased," use "died" or "passed away."
- "Autopsy performed showed multifocal neuronal necrosis, activation of microglia (become CD45 positive), bronchial inflammation..." - Change "become" to "becoming."
Introduction: 3. "COVID-19-associated coinfections represent both a diagnostic and a therapeutic 35 challenge." - There seems to be a number "35" that's not integrated correctly into the sentence. – In our draft manuscript, the number is at its place.
- "As epidemiological restrictions are lifted, respiratory coinfections are expected in 39 the coming winters..." - Consider rephrasing as "As epidemiological restrictions are lifted, respiratory coinfections are expected in the coming winters..." - fixed
- "In January 2022, the term 'flurona' was proposed to describe the coinfection of the 42 influenza virus and SARS‐CoV‐2 [5]." - The numbering seems to be mixed with the text, as "42" appears between "the" and "influenza." - In our draft manuscript, the number is at its place.
- "The continued co‐circulation of SARS‐CoV‐2 and influenza viruses is ex‐ 49 pected to present challenges [7]." - The numbering appears between "ex" and "pected." - In our draft manuscript, the number is at its place.
- "To the best of our knowledge, this is the first paper re- 50 porting an autopsy of a case deceased by flurona." - "re- 50 porting" should be "reporting."- In our draft manuscript, the number is at its place.
Materials and Methods: 8. "Our study was conducted by considering the provisions of the Romanian legal 53 frameworks..." - "legal 53 frameworks" should be "legal frameworks" without the number. -fixed
- "...and Pharmacy, Cluj-Napoca, Romania, No. 63/1.03.2022." - "No. 63/1.03.2022" should be "No. 63/01.03.2022" for consistency. - fixed
- "The tissue samples collected during the autopsy were fixed in formaldehyde 7% for 60 days..." - "formaldehyde 7%" should be "7% formaldehyde." - fixed
- "Paraffin embedding and sectioning were performed 63 using the Tissue-Tek TEC 6 system..." - Change "63 using" to "using" for correct syntax. - fixed
Case Description: 12. "...fever, dysphonia, bilaterally intensified vesicular murmur, and bilateral bronchial 85 rales are observed." - It seems the numbering "85" is embedded in the sentence. - fixed
13."Approximately 5 hours later, the patient displays increased psychomotor agitation, 116 she experiences a sudden desaturation episode..." - The numbering "116" is within the sentence. - fixed
- "...intercostal retractions, Astrup parameters with severe decompensated mixed aci- 119 dosis (hypercapnic, metabolic, lactic) (ph=7.03, pCO2 of 99 mmHg, lac=12mmol/L, 120 HCO3=18 mEq/L)." - The numbering "119 dosis" is not properly separated. - fixed
- "The characteristic plugs of viscous mucus in bronchiolitis (a specific re-action pattern in infants) significantly worsened the progression, likely contributing to 225 the development of spontaneous bilateral tension pneumothorax." - The numbering "225" is misplaced. - fixed
Discussions: 16. "Being rich in B lymphocytes (fig. 3K), we consider that bronchitis/bronchiolitis is 219 likely due to the influenza virus." - Change "219 likely" to "likely 219." - fixed
Conclusions: 17. "Histopathological, 236 tissues injuries caused by both the influenza virus and SARS-CoV-2 can be observed." - "tissues injuries" should be "tissue injuries." - fixed
These are some of the grammatical and English mistakes found throughout the manuscript. Please note that there may be additional errors that were not highlighted in this review.
Thank you very much!
Round 2
Reviewer 3 Report
The manuscript has been significantly improved.